# Investigation of the Optical Properties for Quaternary Se_60−x_Ge_35_Ga_5_Sb*_x_* (*x* = 0, 5, and 10) Chalcogenide Glass

**DOI:** 10.3390/ma15186403

**Published:** 2022-09-15

**Authors:** Huda Allah Abou-Elnour, M. B. S. Osman, M. Fadel, A. M. Shakra

**Affiliations:** 1Physics Department, Faculty of Women for Arts, Science and Education, Ain Shams University, Cairo 11566, Egypt; mbsosman@women.asu.edu.eg; 2Environmental Research Department, National Institute of Occupational Health and Safety (NIOSH), Cairo 2208, Egypt; 3Semiconductor Lab., Physics Department, Faculty of Education, Ain Shams University, Cairo 11341, Egypt; madihafadel@edu.asu.edu.eg (M.F.); amerahsanen@edu.asu.edu.eg (A.M.S.)

**Keywords:** optical properties, chalcogenide glasses, thin films, dispersion energy

## Abstract

A quenching technique was used to prepare the chalcogenide system of the 
${\mathrm{Se}}_{60-x}{\mathrm{Ge}}_{35}{\mathrm{Ga}}_{5}{\mathrm{Sb}}_{x}\;$
(*x* = 0, 5, and 10 at. %), which was deposited as thin films onto glass substrates using a thermal evaporation technique. X-ray diffraction patterns were used for structure examination of the fabricated compositions, which exposes the amorphous nature of the deposited samples. Meanwhile, the chemical compositions of the prepared samples were evaluated and calculated via the energy-dispersive X-ray spectroscopy (EDX), which was in agreement with the measured compositional element percentages of the prepared samples. Based on the optical reflectance *R* and transmittance *T* spectra from the recorded spectrophotometric data ranging from 350 to 2500 nm, the influence of the Sb element on the 
${\mathrm{Se}}_{60-x}{\mathrm{Ge}}_{35}{\mathrm{Ga}}_{5}{\mathrm{Sb}}_{x}$
 thin films’ optical properties was studied. The film thickness and the refractive index were calculated via Swanepoel’s technique from optical transmittance data. It has been observed that the films’ refractive index increases with increasing x value over the spectral range. The refractive index data were used to evaluate the dielectric constants and estimate dispersion parameters 
$E_{o}$
 and 
$E_{d}$
 using the Wemple–DiDomenico model. The optical energy gap 
$E_{g}^{opt}$
 was calculated for the tested compositions. The result of the optical absorption analysis shows the presence of allowed direct and indirect transitions.

## 1. Introduction

Amorphous chalcogenide (ChGs) thin-film compositions have attracted the interest of researchers in recent decades due to their broad and promising modern electronics applications in technological devices [1,2]. Chalcogenide materials have one or more elements, such as Se, S, and Te [1,3,4,5,6,7]. They are of particular interest due to their properties: high transparency in the middle and far-infrared wavelengths, limited optical absorption, high refractive index, reversible phase transformation, insignificant ambient moisture susceptibility, etc. [4,6].

The nonlinear properties of these chalcogenide materials are two or three times greater than ordinary glasses, which makes them attractive for nonlinear optics [1]. Furthermore, they have excellent transmission from visible to far-infrared; their infrared transmission range is between 1 and 16 μm, which mainly covers the two atmospheric windows of 3–5 and 8–12 μm [8]. This is in addition to their good chemical and physical features, which can be used in high-precision molding technology [9]. These properties make this type of material highly recommended for mid- and far-infrared applications [8] such as IR (infrared) lenses and IR detectors and suitable for producing different medical, military, and civil applications [10].

Chalcogenide glasses (ChG) properties can be changed by changing the doping elements, processing techniques, and film deposition methods on the substrate [11]. A number of photo induced phenomena which are supplemented by the variation in optical constants as a shift in the absorption spectrum edge permit these materials to be used for device fabrication due to their high-resolution display and high-density information storage. Thus, we can illustrate the significance of determining optical constants not just to identify their operation but also to develop its application in different technologies as well [11]. These thin-film matrices have also improved various items that can work in infrared optics, such as communication systems and thermal imaging devices. Additionally, they have enhanced remotely sensed distribution systems [1].

Chalcogenides can be obtained in an amorphous or crystalline state. However, they are significant in disordered (amorphous) organizations due to their extraordinary use as the primary material for different optical applications [12]. Among the amorphous ChGs, Selenium and tellurium are the most widely used, but from an applicable view, they are inappropriate in their pure form [13]. Common Se-based materials are favorable [2,6,13] due to commercial and attractive applications [2] because of their high glass-forming ability [13], good thermal stability, and wide transparency window [14]. Contrarily, pure selenium has some disadvantages that can be summarized as low sensitivity, short lifetime [15], and high unitability [6], which limit ChG applications. However, that can be overcome by combining Se with Materials such as Sb, Ga, and Ge [15]. Therefore, adding Ge, Te, Ga, Sb, Cd, etc. as impurities improves its sensitivity, electrical conductivity, and thermal stability giving high crystallization temperature and enhancing the photosensitivity to be comparable with pure Se-material [2,6].

In this study, the Se-Ge-Ga ternary compound was chosen as the base composition because of its adaptable features and infinite industrial and scientific applications. Adding the fourth element Sb, which belongs to the V group, to our system produces a quaternary composition that improves stability, photoconductivity [16], and glass-ability and produces compositional disorder in our compounds [2,13]. Moreover, due to their lower toxicity, antimony-containing systems are preferable for medical applications than compositions with arsenic [14]. This made the Se-Ge-Ga–Sb system an attractive alloy for storage applications and is also known as a potential material for mid-IR fiber [17]. Sb also enhances the optical properties of the Se-Ge-Ga system; It reduces 
$E_{g}\;$
 (the optical band-gap energy) [13]. Germanium was chosen as one of the matrix components because it adapts and strengthens the system’s average bond, thus increasing the glass formation area. As a result, an incredibly dense glassy matrix is formed due to its size and electronegativity, with compatible host values [13].

The results of earlier investigations revealed that thin films had not been used before to explore the impact of changing Sb percent on Se-Ge-Ga. To the best of our knowledge, there are some studies for one of these compositions [8,18] but the investigation of the effect of changing Sb on different compositions of Se-Ge-Ga was very rare.

In our study, the optical characterization of the vacuum evaporated Se_60−*x*_Ge_35_Ga _5_Sb*_x_* thin-film system (where *x* = 0, 5 and 10%) is reported. The optical constants are assumed and studied by examining transmission ranges in 350–2500 nm wavelengths. The Parameters of Wemple and Di-Domenico [19,20], in addition to dielectric constants and the relationship between the parameters obtained, are investigated and discussed. Our results were in agreement with previous chalcogenide glass studies [21,22].

## 2. Materials and Methods

Melt-quenching techniques were used to synthesize bulk Se_60−*x*_Ge_35_Ga_5_Sb*_x_* (*x* = 0, 5, 10 at. %) alloys from high-purity components of Ge, Se, Ga, and Sb (5N, Sigma Aldrich, St. Louis, MO, USA). First, each element was weighted using a sensitive electrical balance according to their stoichiometric ratio (a 5-g total weight). Then, the mixture was poured into pre-cleaned evacuated silica ampoules and closed in a 10^−5^ Torr vacuum. Each tube was then separately put in a rocking and a wobbling furnace. The furnace temperature was elevated to 50 °C (melting point (MP) of Ga ≈ 30 °C), which was kept constant for one hour. Then, the temperature was raised gradually at a rate of 50 °C/h until 220 °C (≈MP of Se) and kept stable for two hours, followed by a rising progressively at the same rate up to 640 °C (≈MP of Sb) and kept for another two hours (for the compounds doped with Sb). The temperature was finally increased to 950 °C (MP of Ge ≈ 940 °C) and remained constant for around 20 h. (The examined compound’s homogeneity and quality are ensured by the long synthesis time and continuous mechanical shaking of the mixture in the vibrating furnace). Afterward, the molten compounds inside each ampoule were quenched with icy water to obtain studied specimens in the amorphous state.

The fine powder obtained by crushing the bulk ingot was used for evaporation. Thin films of the prepared compositions were deposited on a pre-cleaned glass substrate using the thermal evaporation technique at a high vacuum coating unit (Type Edwards E306 A) at a vacuum of 10^−5^ Torr. The substrates were fixed on a suitable rotatable holder approximately 20 cm above the evaporator unit to generate homogeneous deposited films on a plane substrate. During the deposition procedure, the substrate temperature was kept at room temperature.

A scanning electron microscope (Jeol JSM 5400, Tokyo, Japan) with an EDX (Energy-dispersive X-ray) detector was used to determine the elemental composition of the as-deposited films at a 30 kV accelerating voltage. Simultaneously, a Philips X-ray diffractometer, Amsterdam, Netherlands) provided with a copper target studied the structure of the alloys. A Ni filter at 40 kV and 20 mA was used for X-ray diffraction (XRD) analysis of the investigated samples.

Transmission and reflection of the optical spectrum for the evaporated films were measured at normal incidence with unpolarized light at room temperature in the wavelength range 350–2500 nm by (Jasco, model V570, Reroll-00, UV–VIS-NIR, Tokyo, Japan) a calibrated double beam spectrophotometer.

## 3. Results and Discussion

### 3.1. Structure Identification of 
$Se_{60-x}Ge_{35}Ga_{5}Sb_{x}$
 Thin Films

#### 3.1.1. Electron Dispersive X-ray Spectroscopy (EDX)

Figure 1 represents the EDX analysis of the studied compositions (measured at three randomly selected regions), the surface morphology of 
${\mathrm{Se}}_{60-x}{\mathrm{Ge}}_{35}{\mathrm{Ga}}_{5}{\mathrm{Sb}}_{x}$
 at (*x* = 0, 5, 10 at. %) samples obtained from the scanning electron microscope can be found in Appendix A. The elemental analysis of these specimens leads to the chemical formula Se_60.59_Ge_32.20_Ga_7.21_, Se_53.12_Ge_37.80_Ga_6.01_Sb_3.07_, and Se_50.88_Ge_36.85_Ga_5.47_Sb_6.80_, revealing the percentages of the constituent elements in the calculated compositions are nearly similar to those in the laboratory-prepared ingots (see Table 1). Furthermore, it confirmed the absence of any odd component in the investigated samples.

#### 3.1.2. X-ray Diffraction (XRD) Characterization

Figure 2 illustrates the XRD pattern for the thermally evaporated 
${\mathrm{Se}}_{60-x}{\mathrm{Ge}}_{35}{\mathrm{Ga}}_{5}{\mathrm{Sb}}_{x}$
 (*x* = 0, 5, and 10 at. %) thin-film samples. Analysis of this pattern shows an amorphous phase. A broad hump can be observed from charts of each composition, and no sharp peaks were observed. This can be explained by the vaporized molecules randomly precipitating on the substrate surface during evaporation.

### 3.2. Optical Properties of Se_60−x_Ge_35_Ga_5_Sb_x_ (x = 0, 5, 10 at. %) Thin-Films

The optical absorption of examined compositions, especially the absorption edge, is considered essential for understanding the electronic nature of the specimens [23,24]. Swanepoel’s technique [25] calculated the optical constants of the studied films along with the spectral distribution.

#### 3.2.1. Transmittance and Reflectance of the Spectral Distribution for Studied Films

The variation of the transmittance *T*(*λ*) and reflectance *R*(*λ*) with wavelength *λ* for the examined films at almost the same thickness (450 nm) were obtained at normal incidence using a dual-beam spectrophotometer of wavelength range (350–2500 nm) at room temperature (303 K), as illustrated in Figure 3a,b. As noticed in these figures, films become transparent at longer wavelengths (1900–2500 nm), *T + R* ≈ 1, which indicates that the films become transparent with no absorption or scattering occurring; extinction coefficient *k* ≈ 0 [24,26].

Comparative transmittance spectra for films with various values of *x* are noticed in Figure 3a. By increasing the antimony (Sb), a redshift of the interference-free region was observed with fringes position changes at low energy; this redshift in film transmission proves that in the Urbach tail, light can form mobile carriers (holes) in films below the optical absorption edge [27].

Figure 4 illustrates the relationship between *T*, *R*% along 
λ(nm)
 for Se_50_Ge_35_Ga_5_Sb_10_ as an example for all examined films. The maxima of the transmission spectra at nearly the same wavelength as the reflectance spectrum’s minim and vice versa. This indicates that the deposited films are optically homogeneous [23,28]. The well-known Swanepoel technique [25] was applied in our work. This method relies only on the transmission spectrum’s extreme interference fringes to determine the optical properties of the deposited Se_60−*x*_ Ge_35_Ga_5_Sb*_x_* (*x* = 0, 5, 10 at. %) films.

#### 3.2.2. Estimation of the film thickness and the refractive index *n*

Swanepoel’s method [25], considering the idea of Manifacier et al. [23], can be used for thickness and refractive index determination for the examined films from the measured transmission spectrum, built on the creating envelopes of *T_m_* and *T_M_* around the interference minima and maxima of the transmittance spectrum. The initial refractive index 
$n_{i}$
 values can be obtained by knowing the tangent points of the transmittance spectrum and the envelopes by the following formula [25]:
(1)
ni=[M+(M2−S2)12]

in which

(2)
$$M=2S\;\frac{T_{M}-T_{m}}{T_{M}T_{m}}+\frac{s^{2}+1}{2}$$

where the values of 
$T_{M}$
 and 
$T_{m}$
 are the tangent point of the envelopes at wavelengths, when the experimental transmission spectrum and the upper and lower envelopes are tangent, and the substrate’s refractive index *S* = 1.5.

The initial approximation of film thickness can be derived using the formula [25]:
(3)
$$t_{i}=\frac{\lambda_{1}\lambda_{2}}{2\left(\lambda_{1}n_{2}-\lambda_{2}n_{1}\right)}$$

where 
$n_{1}$
 and 
$n_{2}$
 are the two adjacent maxima (or minima) refractive indices at wavelengths 
$\lambda_{1}$
 and 
$\lambda_{2}$
, respectively. Then, the initial film thickness 
$\bar{t_{i}}$
 values were calculated. These parameters are needed to calculate 
$m_{o}$
 “the order number of extremes” together with 
$n_{i}$
 by the interference fringes basic equation 
$2n_{i}\bar{t_{i}}=m_{o}\lambda$
. Then, by taking the corresponding exact integer or half-integer value of 
$m_{f}$
, the accuracy of the film thickness is significantly improved. That, in this manner, has less dispersion, and its average value is used to calculate the final thickness of the film 
$\bar{t_{f}}$
 which is estimated to represent the final thickness for each composition. Then, finally, the final value of the refractive index 
n
 can be determined by the average value of 
$\bar{t_{f}}$
, together with the exact value of 
$m_{f}$
 from formula 
$2n\bar{t_{f}}=m_{f}\lambda$
.

The values of *n* for all investigated film compositions can be fitted using a two-term Cauchy equation [29]: 
$n=a+b/\;\lambda^{2}$
, as presented in the Figure 5 inset, which can be used in the extrapolation at a shorter wavelength. The obtained values of 
a
 and *b* for the whole range of 
λ
 are summarized in Table 2. The deposited films’ experimentally calculated refractive index *n* has a spectrum variation that matches the Cauchy relation’s estimated for various Sb concentrations (See Figure 5).

From Figure 5, The refractive index *n* has higher values at low wavelengths in the spectral area (<670 nm) known as the strong absorption region. Then, it reduces as the wavelength increases, becoming relatively flat above 1500 nm. Furthermore, increasing Sb concentration increases the refractive index *n* at any wavelength value. This increase can be discussed by the great polarizability of bigger Sb atoms (the atomic radius, 1.45 A°) compared with that of Se atoms (1.14 A°) [30,31]. This behavior is in agreement with that of [31,32].

#### 3.2.3. Estimation of the Optical Constants *k*

Knowing the film thickness *t* and refractive index *n* values, one can estimate the absorption coefficient *α* by calculating the absorbance *x* in the fundamental edge region with the equation considered by Swanepoel [25] as follows:
(4)
$$\;\alpha =\frac{1}{t}ln(x^{-1})$$


In the case of the strong absorption region where the minima and maxima interference merge to a single curve 
$T_{o}$
, the absorbance *x* was given by:
(5)
$$x=\frac{{\left(n+1\right)}^{3}\left(n+s^{2}\right)}{16n^{2}s\;}T_{o}$$
In the case of the weak and medium absorption region, the absorbance *x* is calculated in terms of the interference extremes by the following equation:
(6)
$$x=\frac{E_{M}-{\left[E_{M}^{2}-{\left(n^{2}-1\right)}^{3}\left(n^{2}-s^{4}\right)\right]}^{\frac{1}{2}}}{{\left(n-1\right)}^{3}\left(n-s^{2}\right)}\;$$

where 
$E_{M}=\frac{8n^{2}s}{T_{M}}+\left(n^{2}-1\right)\left(n^{2}-s^{2}\right)$
.

For films under examination, the extinction coefficient *k* was determined knowing the value of 
α
 with the wavelength 
λ
 by the following relation:
(7)
$$k=\frac{\alpha \lambda }{4\pi }\;$$


The relationship between the extinction coefficient *k* and wavelength 
λ
 was represented in Figure 6.

#### 3.2.4. The Analysis of Absorption Coefficient and the Calculation of Optical Energy Gap 
$E_{g}^{opt}$


The optical absorption study in materials gives a simple way to explain some of the band structure characteristics and energy gaps of non-metallic material [1]. Based on the value of 
α 
 determined by the Swanepoel method, the absorption edge can be divided into two absorption regions depending on its energy (low and high) for many amorphous materials [24,26]. The dependency of *α* on photon energy *hν* for the studied films is presented in Figure 7.

Depending on the absorption coefficient values, the absorption edge is split into two regions: the 1st region is for lower absorption coefficients *α* < 10^4^ cm^−1^, where absorption at lower photon energies usually follows the Urbach’s tail rule [33]. In this region, the absorption is due to transitions between localized states in the exponential tail in one band and extended state of the other band as displayed in the following formula:
(8)
$$\alpha \left(h\nu \right)=\alpha_{\circ }exp\left(h\nu /E_{e}\right)$$

where 
$\alpha_{\circ }$
 is constant and *E_e_* is the Urbach’s energy that represents the disorder degree in amorphous semiconductors [34,35] and is explained as the width of the tail of localized states in the optical energy gap.

Figure 7 illustrates the relation of 
lnα
 as a function of *hυ*. The *E_e_* and 
$\alpha_{\circ }$
 values can be deduced from the first region’s least-square fitting in compliance with Equation (8) and listed in Table 3. It is noticed that by increasing the Sb percentage, the localized state tail’ width in the band gap increases due to a rise in Urbach energy *E_e_* values. It could be explained by an increased disorder in the specimens that included Sb elements compared to other specimens, which may be attributed to the formation of homopolar bonds with the addition of Sb [32].

While the 2nd region is for higher absorption coefficients 
$\alpha \geq 10^{4}\;$
 cm^−1^_,_ this relates to transitions between extended states in both conduction and valence bands according to Tauc’s law [36,37], at which the optical absorption edge follows the next equation:
(9)
$$\alpha h\nu =A{\left(h\nu -E_{g}^{opt}\right)}^{r}$$


The film quality is represented by the edge width parameter A, computed from the linear section of Equation (9). 
$E_{g}^{opt}$
 is the material’s optical energy gap, and *r* is an index parameter that identifies the optical transition’s type and theoretically may be equal to 2, 
$\frac{1}{2}\;$
 for allowed indirect and direct transmission while equivalent to 3, and 
$\frac{3}{2}\;$
 for forbidden indirect and direct transition, respectively. Plotting 
${\left(\alpha h\upsilon \right)}^{1/r}$
 versus 
hυ 
 according to Equation (9) is the best way to know the optical transition type and estimate the relevant energy value. The intercept on the photon energy axis gives the optical band gap. Figure 8 illustrates that the plot of (
$\alpha h\nu {)}^{1/2}$
 vs. 
hν
 shows a straight line indicating the existence of allowed optical indirect transition 
$E_{g}^{ind}$
 values, while the relation of 
${\left(\alpha h\nu \right)}^{2}$
 against 
hν
 gives a straight line indicating the presence of allowed direct transitions 
$E_{g}^{dir}$
 for all the investigated samples as shown in Figure 9. The energy gaps 
$E_{g}^{dir}$
 and 
$E_{g}^{ind}\;$
 values were obtained by intersecting the photon energy axis 
hν
 with the intercept of the extrapolation of the linear part to zero absorption and recorded in Table 3.

It is observed from Table 3 that with increasing Sb content, the estimated optical energy gap *E_g_* in direct and indirect transitions decreases while the Urbach energy *E_e_* rises. This agrees with the effect of Sb on chalcogenide films in References [31,32], which might be due to the higher compositional disorder of chalcogenide glasses [26].

The presence of Sb induces localized states in the band gap, which causes the band edges’ tailing [31]. Increased band tailing in the gap can explain the reduction in 
$E_{g}$
 in amorphous films [38].

The band edges of amorphous materials are generally broadened by the lake of long-range order and by the presence of defects, which is probably due to the structural defects formation. As a result, the localized states at/or near the band edges increase the system’s disorder and the band tail width [26]. In other words, it may be explained that unsaturated Sb atoms could form a defect center as the amount of Sb increases. This could cause perturbation (system disorder) and broaden the mobility gap’s valence and conduction band edges [39].

According to Nagel et al. [40], the optical band gap variation can also be explained in terms of the system’s average bond energy with Sb incorporation. The value of the optical band gap 
$E_{g}$
 decreases as the width of the localized state 
$E_{e}$
 increases. This may be attributed to increasing the amount of Sb, causing the Sb–Sb bonds formation (bond energy = 30.22 kcal/mol [41]) and Sb–Se bond, which decreases the Se–Se bond concentration and leading to the 
$E_{e}$
 increment (decrease of 
$E_{g}$
) [42] (see the bond energy Table 4 [41]).

#### 3.2.5. Determination of the Dielectric Constants 
$(\varepsilon_{L}$
, 
$\varepsilon_{\infty }$
) at High Frequency

The refractive index (*n*) data can be used to get the high-frequency dielectric constant *ɛ* by two different methods [43].

The first method explains the contribution of free carriers and dispersion’s lattice vibrational mode to the real dielectric constant (
$\varepsilon_{1})$
. It is widely recognized that the electronic transition in a given material is extra immediately related to the complex dielectric constant *ε* = (*
$\varepsilon_{1}-i\varepsilon_{2}$
) instead of the complex refractive index, 
N=n−ik
. According to this method, the contribution of free carriers to the real part of the dielectric constant (
$\varepsilon_{1}$
) can be described by References [23,26] as:
(10)
$$\varepsilon_{1}=n^{2}-k^{2}=\varepsilon_{L}-\beta \lambda^{2}$$

where 
$\beta =\frac{e^{2}N\;}{4\pi^{2}C^{2}\varepsilon_{o}m^{*}}$
 and 
$N/m^{*}=\frac{\varepsilon_{o}\varepsilon_{L}}{e^{2}}\omega_{p}^{2}$
 and where *c* is the light velocity, 
$\varepsilon_{o}$
 is the free space dielectric constant, *e* is the electronic charge,
$\;\varepsilon_{L}$
 is the lattice dielectric constant, 
$N/m^{*}$
 is the ratio of the free charge carrier’s density concentration to its effective mass, 
$\omega_{p}$
 is the plasma frequency (where ω is the incident light frequency) and can be determined by the ratio:
(11)
$$\omega_{p}^{2}=\frac{e^{2}N/m^{*}}{\varepsilon_{o}\varepsilon_{L}}$$


The imaginary part is 
$\varepsilon_{2}=2nk$
, in the transparent (non-absorbing) region (*k* ≈ 0); the dielectric constant is attributed to free and bounded electrons (
$\varepsilon_{1}=n^{2}$
). The variation of real part 
$\varepsilon_{1}$
 versus 
$\lambda^{2}$
 was depicted in Figure 10. Thus, the relation gives a straight line according to Equation (11). The 
$\varepsilon_{L}$
 and 
$N/m^{*}$
 values for the films were obtained from the vertical axis intercept and slope, and the obtained values are used to determine the plasma frequency values 
$\omega_{p}$
 listed in Table 5.

However, the second method depends on the dispersion arising from the bound carriers in an empty lattice. The high-frequency properties of the synthesized films are expressed as a single oscillator of wavelength *λ**_o_* at high frequency by applying a simple classical dispersion relation [44]. If n_o_ is the refractive index of an empty lattice at infinite wavelength *λ_o_*, it will vary as follows:
(12)
$$\frac{\left(n_{o}^{2}-1\right)}{\left(n^{2}-1\right)}=1-{\left(\lambda_{o}/\lambda \right)}^{2}$$

where *λ**_o_* and *n**_o_* have been calculated from plots of (*n^2^* − 1)^−1^ against *λ*^−2^ given in Figure 11. The values of 
$n_{o}^{2}$
 were obtained by extrapolating the obtained lines to the y-axis, while *λ**_o_* was determined from the slopes; values *ε*_ꝏ_; *(ε*_ꝏ_ = 
$n_{o}^{2}$
*)* and *λ**_o_* are illustrated in Table 5.

The data recorded in Table 5 clarified that both 
$\omega_{p}$
 and 
$N/m^{*}$
 increase with increasing Sb content. This can be due to the variety of film stoichiometry. It was also observed that there is a slight difference between the computed values of 
$\varepsilon_{L}\;$
 and 
$\varepsilon_{\infty }$
; this difference might be attributed to the free carriers’ contribution.

#### 3.2.6. Wemple–DiDomenico Model for Dispersion Energy Parameters

The Wimple–DiDomenico (WDD) model [44,45] is based on the single-oscillator technique below the band gap in the range from visible to near IR region; it depends on the recorded refractive index data and has the following form:
(13)
$$n^{2}-1=\frac{E_{o}E_{d}}{E_{o}^{2}-{\left(h\nu \right)}^{2}}$$

where *E**_o_* is the single oscillator energy, and 
$E_{d}$
 is the dispersion energy, which acts as a measure for the inter-band optical transitions strength [34]. These dispersion parameters are highly valuable for basic empirical laws that apply to a large set of semiconducting materials [19], where dispersion is considered a significant influence in choosing the optical material because it is an essential factor in designing equipment for spectrum dispersion and optical communication [34].

Plotting 
${\left(n^{2}-1\right)}^{-1}$
 versus 
${\left(h\nu \right)}^{2}$
 gives a straight line for the normal behavior having the slope (
$E_{o}E_{d}{)}^{-1}$
 and its intercept (
$E_{o}/E_{d}$
). The 
$E_{o}$
 and 
$E_{d}$
 values were obtained for the investigated specimens by fitting the straight lines to the smaller energy data, as shown in Figure 12 and listed in Table 6. The 
$E_{d}$
 and 
$E_{o}$
 parameters helped to calculate the static refractive index 
$n_{s}\left(0\right)$
 at *hν*
→
0, the static high-frequency dielectric constant 
$\varepsilon_{s}$
, and the Wimple–DiDomenico 
$E_{g}^{WD}$
 band gab.
Figure 12Plots of (*n*^2^ − 1)^−1^ versus (*hυ*)^2^ for Se_60−*x*_Ge_35_Ga_5_Sb*_x_* thin films.
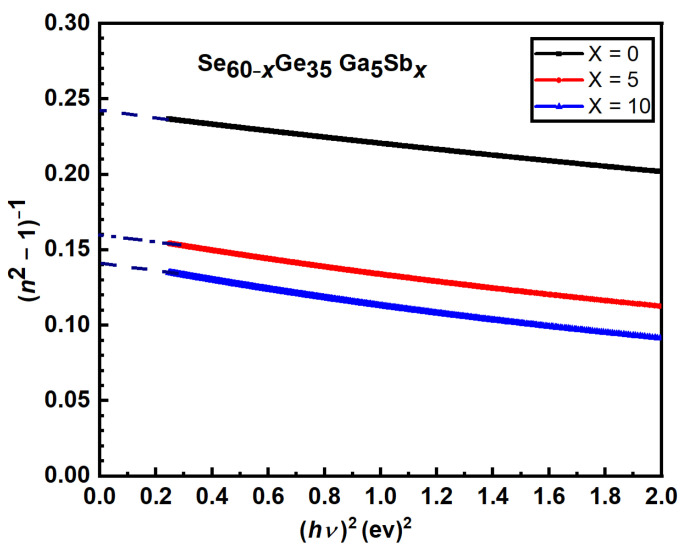

materials-15-06403-t006_Table 6Table 6Values of the Wemple–DiDomenico oscillator parameters for Se_60−*x*_Ge_35_Ga_5_Sb*_x_* (*x* = 0, 5, and 10) films.Film Composition

Ed(eV)



Eo(eV)



ns(0)


***ε_s_***


EgWD (eV)


**Se_60_Ge_35_Ga_5_**
17.426 ± 0.06933.375 ± 0.09052.486.161.687
**Se_55_Ge_35_Ga_5_Sb_5_**
18.507 ± 0.06962.874 ± 0.08802.77.4391.437
**Se_50_Ge_35_Ga_5_Sb_10_**
18.747 ± 0.06862.564 ± 0.08732.8828.311.282
where 
$n_{s}\left(0\right)=\sqrt{1+\frac{E_{d}}{E_{o}}}$
, 
$\varepsilon_{s}={\left(n_{s}\left(0\right)\right)}^{2}$
 and 
$E_{g}^{WD}=\frac{Eo}{2}$
 [44].

The assessed values of 
$E_{o}$
, 
$E_{d}$
,
$\;n_{s}\left(0\right)$
, 
$\varepsilon_{s}$
, and 
$E_{g}^{WD}$
 are listed in Table 5 as a function of Sb content.

Table 6 clarified that both the oscillator strength (dispersion energy) 
$E_{d}$
 and the static high-frequency dielectric constant 
$n_{s}\left(0\right)$
 increase while single-oscillator energy 
$E_{o}$
 decreases with Sb content increment. Increasing the excess of the Se–Se covalent bond could explain the increment in 
$E_{d}$
 with rising Sb concentration. It should be noted that, with more Sb atoms added, a modification to the homopolar Se–Se bond occurs [32].

The decrease of the 
$E_{g}^{opt}$
 by increasing Sb content can be discussed by the strong bond between Se and Sb atoms. The Se atoms also fill the Ge’s available valences; after all of these bonds, there are still unsatisfied Se valences that have been formed and can be satisfied when Se–Se bonds are formed. As a result, by increasing Sb content, the excess Se–Se bonds are produced [32].

The average *E_o_* gap provides quantitative information about the material’s overall band structure. In detail, the *E_o_* parameter refers to the distance between the valence and conduction bands’ centers of gravity, identified as the WDD gap. Thus, it is associated with the systems’ average bond strength or cohesive energy.

## 4. Conclusions

Thermal evaporation technique was used to synthesize thin 
${\mathrm{Se}}_{60-x}{\mathrm{Ge}}_{35}{\mathrm{Ga}}_{5}{\mathrm{Sb}}_{x}\;$
 (*x* = 0, 5, 10 at. %) films. X-ray analysis specifies that the resulting films are amorphous for all the examined compositions. Swanepoel’s method was used to study the optical characterization of the synthesized thin films. A two-term Cauchy dispersion relationship fitted the refractive index of these films. The optical constants (*n* and *k*), the optical energy gap, and the width of localized states were calculated using spectral transmittance in the range 350 nm to 2500 nm.

The optical absorption analysis confirmed that the absorption mechanism is due to allowing direct and indirect transitions. The optical energy gap 
$E_{g}^{opt}$
 decreases with increasing Sb content. This decrease in the 
$E_{g}^{opt}$
 may be attributed to the disorder increase of the prepared film samples. The dispersion parameters (*E_d_*, *Eₒ*, 
$\varepsilon_{L}$
, and *N/m**) were obtained by analyzing the refractive index data using the single oscillator model. The dispersion energy 
$\;E_{d}\;$
 values are found to increase, whereas a decrease was found in 
$E_{ο}\;$
 values, with an increase in Sb-content.

By increasing the dopant (Sb) content, it was observed that the absorption coefficient increases. The *α* increase refers to the fact that the charge carriers absorb more energy. The great value of the absorption coefficient and other related parameters make these materials appropriate for optical data storage.

## Figures and Tables

**Figure 1 materials-15-06403-f001:**
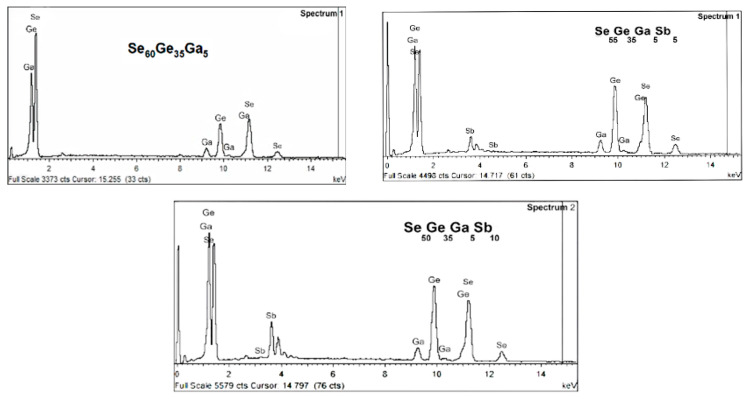
EDX spectra of 
${\mathrm{Se}}_{60-x}{\mathrm{Ge}}_{35}{\mathrm{Ga}}_{5}{\mathrm{Sb}}_{x}$
 at (*x* = 0, 5, 10 at. %) samples.

**Figure 2 materials-15-06403-f002:**
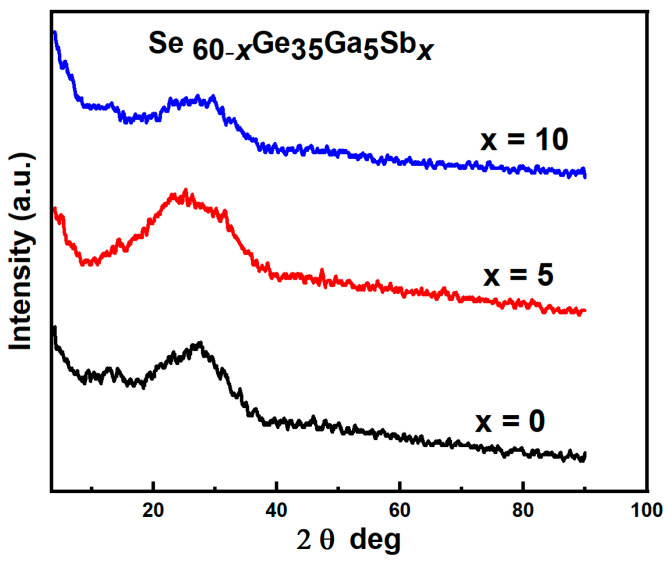
XRD patterns for 
${\mathrm{Se}}_{60-x}{\mathrm{Ge}}_{35}{\mathrm{Ga}}_{5}{\mathrm{Sb}}_{x}\;$
(*x* = 0, 5, and 10 at. %) thin films.

**Figure 3 materials-15-06403-f003:**
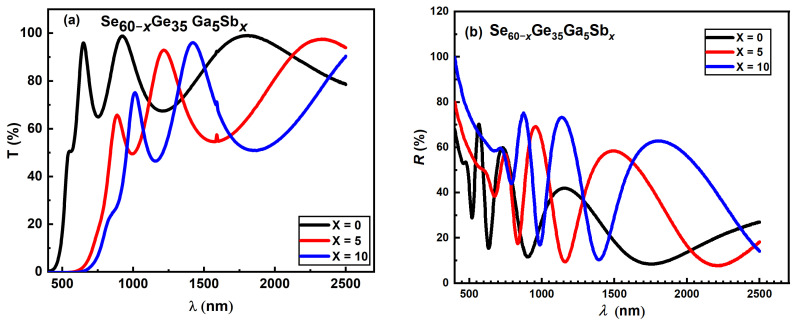
(**a**,**b**) The transmittance and reflectance spectra of the evaporated film at a thickness of 450 nm.

**Figure 4 materials-15-06403-f004:**
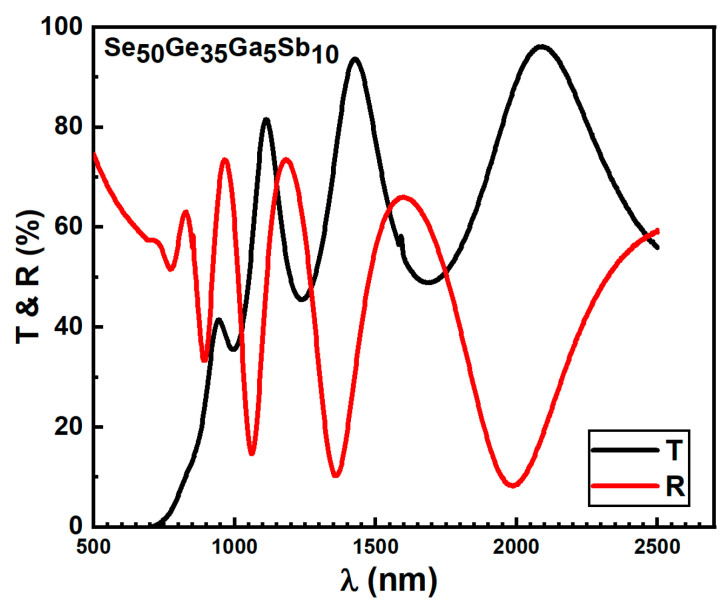
Transmittance and reflectance spectra of Se_50_Ge_35_Ga_5_Sb_10_ film.

**Figure 5 materials-15-06403-f005:**
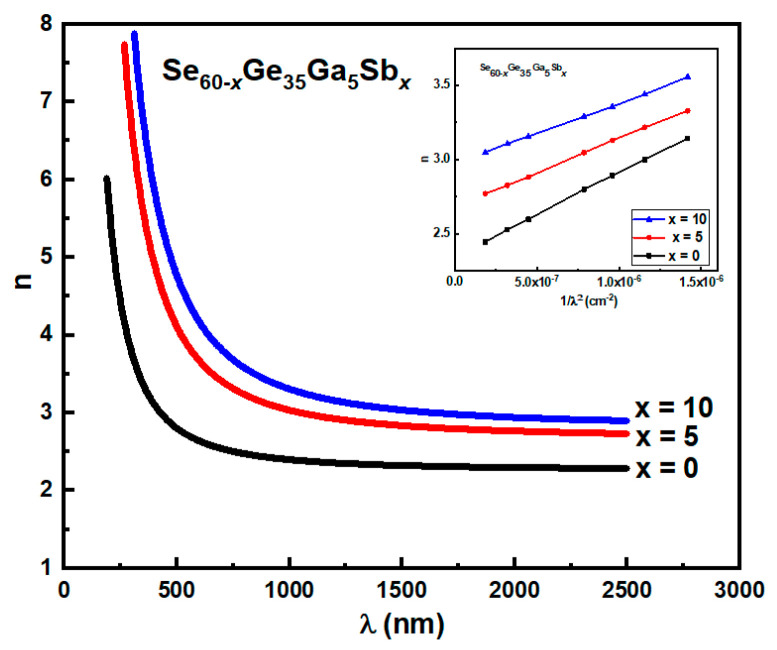
The refractive index’s spectral distribution along with the wavelength for studied films. Inset: The Cauchy equation fitting for the as-deposited films.

**Figure 6 materials-15-06403-f006:**
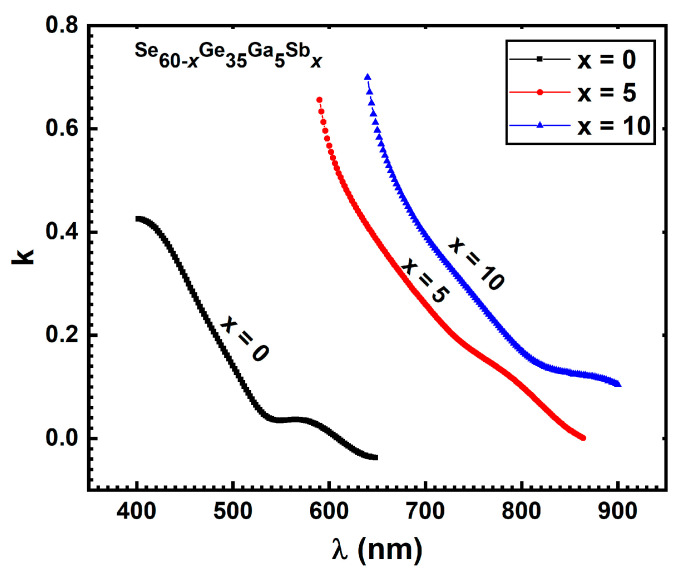
The dependency of extinction coefficient *k* on the wavelength 
λ
 for the as-deposited films.

**Figure 7 materials-15-06403-f007:**
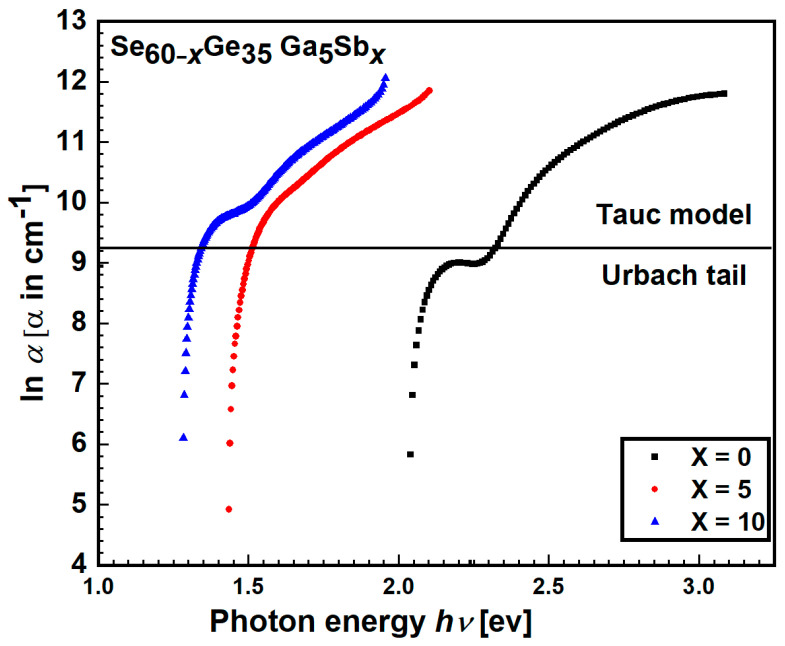
Photon energy 
hν
 versus ln(α).

**Figure 8 materials-15-06403-f008:**
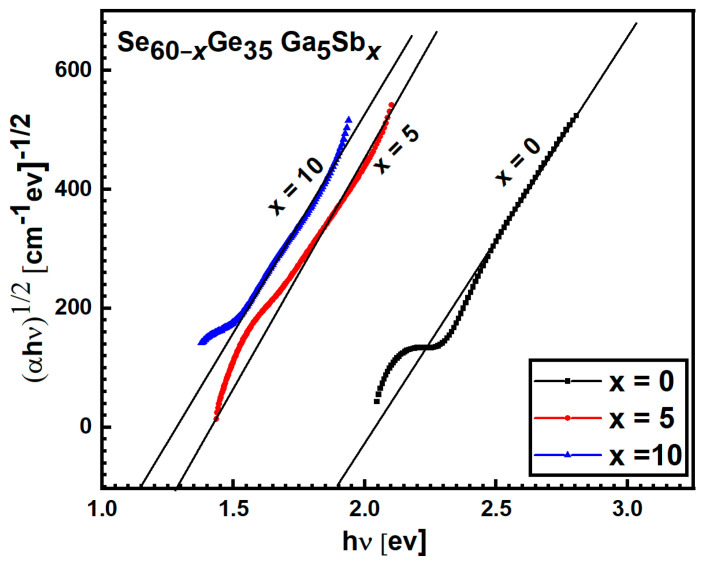
The relation of (
$\alpha h\nu {)}^{1/2}$
 with respect to 
hν
 for Se_60−*x*_Ge_35_Ga_5_Sb*_x_* (*x* = 0, 5, 10) films.

**Figure 9 materials-15-06403-f009:**
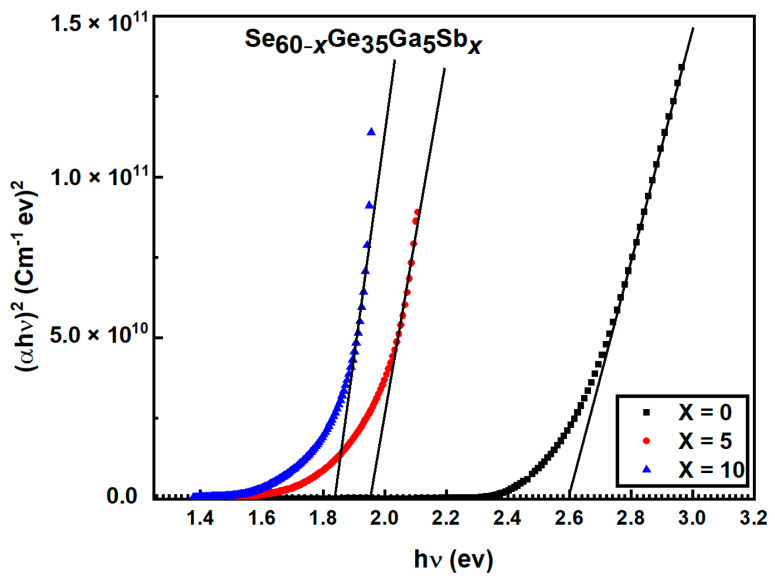
The relation of (
$\alpha h\nu {)}^{2}$
 with respect to 
hν
 for Se_60−*x*_Ge_35_Ga_5_Sb*_x_* (*x* = 0, 5, 10) films.

**Figure 10 materials-15-06403-f010:**
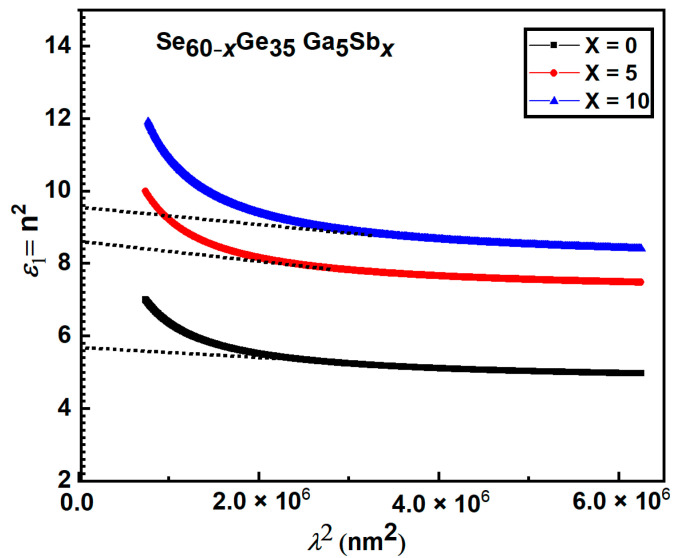
Plots of 
$\varepsilon_{1}$
 against 
$\lambda^{2}$
 for Se_60−x_Ge_35_Ga_5_Sb*_x_* thin films.

**Figure 11 materials-15-06403-f011:**
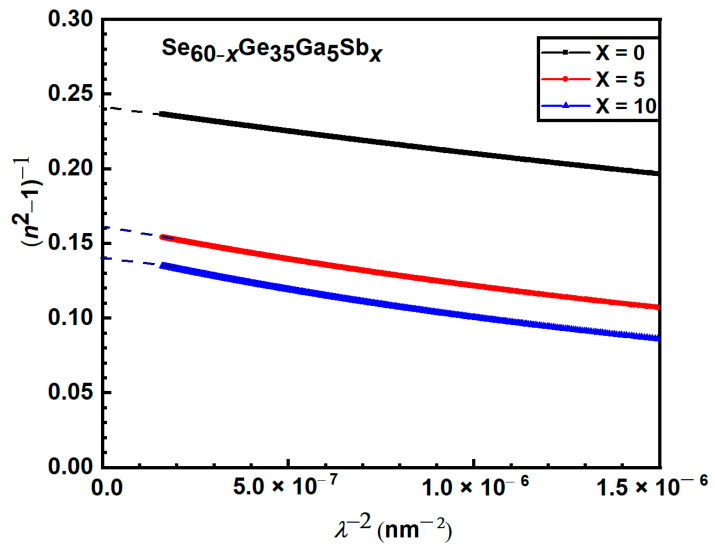
Plots of 
${\left(n^{2}-1\right)}^{-1}$
 against *λ*^−2^ for Se_60−*x*_Ge_35_Ga_5_Sb*_x_* thin film.

**Table 1 materials-15-06403-t001:** The average at. % of the constituent elements obtained from EDX of the studied 
${\mathrm{Se}}_{60-x}{\mathrm{Ge}}_{35}{\mathrm{Ga}}_{5}{\mathrm{Sb}}_{x}$
 compositions.

*X*	Se (at. %)	Ge (at. %)	Ga (at. %)	Sb (at. %)	The Actual Formula
0	60.59	32.20	7.21	0	${\mathrm{Se}}_{60.59}{\mathrm{Ge}}_{32.20}{\mathrm{Ga}}_{7.21}$
5	53.12	37.80	6.01	3.07	${\mathrm{Se}}_{53.12}{\mathrm{Ge}}_{37.80}{\mathrm{Ga}}_{6.01}{\mathrm{Sb}}_{3.07}$
10	50.88	36.85	5.47	6.80	${\mathrm{Se}}_{50.88}{\mathrm{Ge}}_{36.85}{\mathrm{Ga}}_{5.47}{\mathrm{Sb}}_{6.80}$

**Table 2 materials-15-06403-t002:** The constant values of *a* and *b* according to Cauchy-relation for the studied samples.

Compositions	a	*b*
$Se_{60}Ge_{35}Ga_{5}$	2.1726	$3.50762\;\times \;10^{5}$
$Se_{55}Ge_{35}Ga_{5}Sb_{5}$	2.6897	$3.40036\;\times \;10^{5}$
$Se_{50}Ge_{35}Ga_{5}Sb_{10}$	3.0124	$3.41857\;\times \;10^{5}$

**Table 3 materials-15-06403-t003:** The determined values of 
$E_{e}$
, 
$\alpha_{ο}.E_{g}^{dir}$
, 
$E_{g}^{ind}\;$
 for Se_60−_*_x_*Ge_35_Ga_5_Sb*_x_* (*x* = 0, 5, 10) films.

Film Composition	$E_{e}\;(m.\mathrm{eV})$	$\alpha_{ο}$ $\left({\mathrm{cm}}^{-1}\right)$	$E_{g}^{dir}\;(\mathrm{eV})$	$E_{g}^{ind}\;(\mathrm{eV})$
$Se_{60}Ge_{35}Ga_{5}$	21.3	1.49 × $10^{-39}$	2.6	1.85
$Se_{55}Ge_{35}Ga_{5}Sb_{5}$	30	2.47 × $10^{-18}$	1.95	1.4
$Se_{50}Ge_{35}Ga_{5}Sb_{10}$	37	1.37 × $10^{-12}$	1.85	1.2

**Table 4 materials-15-06403-t004:** Bond energies in Se-Ge-Sb elements of different bonds.

Bond	Bond Energy (kcal/mol)
Sb–Sb	30.22
Ge–Sb	33.76
Ge–Ge	37.60
Se–Se	44.04
Sb–Se	43.98
Ge–Se	49.44

**Table 5 materials-15-06403-t005:** Calculated values of the optical parameters 
$\varepsilon_{L}$
, *ε*_ꝏ_, 
$\omega_{p}$
, and *λ_o_* for the prepared films.

Film Composition	$\varepsilon_{L}$	$\varepsilon_{\infty }$	$N/m^{*}$ $\left(m^{-3}kg^{-1}\right)$	$\omega_{p}$ $s^{-1}$	*λ*_o_ (nm)
$Se_{60}Ge_{35}Ga_{5}$	5.71 ± 0.0980	5.166 ± 0.00012	1.227 × $\;10^{56}$	2.49 × $\;10^{14}$	446.65
$Se_{55}Ge_{35}Ga_{5}Sb_{5}$	8.69 ± 0.1647	7.273 ± 0.000065	2.454 × $10^{56}$	2.85 × $\;10^{14}$	492.16
$Se_{50}Ge_{35}Ga_{5}Sb_{10}$	9.85 ± 0.1647	8.127 ± 0.000059	3.681 × $10^{56}$	3.28 × $\;10^{14}$	534.88

## Data Availability

The raw data required to reproduce these results cannot be shared at this time as the data also form part of an ongoing study.

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
