# Peer review of "Investigation of the Optical Properties for Quaternary Se60−xGe35Ga5Sbx (x = 0, 5, and 10) Chalcogenide Glass"

_materials, 2022, doi:10.3390/ma15186403_

Round 1

Reviewer 1 Report

The paper describes the preparation of a quaternary Chalcogenide glass, SeGeGaSb, and the characterization of its optical properties. Such a system may be of interest due to its simple technology of thin film preparation and optical properties in the infrared spectral regime.

A clear description of the experimental part is given. Still, the optical properties of the quaternary SeGeGaSb system in the infrared spectrum have been investigated by several researchers in recent years. Hence, the authors need to elaborate more clearly on what is new in their study. Also, provide discussion on how their film compares to SeGeGaSb films prepared by others.

The grammar and the general quality of the English writing need to be improved.

Minor details

L97. Incorrect spelling, Joel instead of Jeol.

Inconsistency in letter style: symbols in text are often written in roman style but with italic style in equations (in general, only variables should be written with italic style).

Text in Figure caption often starts with a small (not capital) letter

Figure 8 is redundant (the same results are shown very clearly in Figure 7)

Some strange sentences, seen at a glance, need to be re-structured/changed, f.ex.  L54 Fortunately …,  L240 That can be .., L249 Characterizes the …, L329 The Wimple …,  L341 The parameters …

The sentence starting in L263 needs to be fragmented.

Reviewer 2 Report

This paper is interesting and can be of great help to researchers who are working on similar materials. However, I have some recommendations that should be considered by the authors before the paper is accepted for publication:

1) There are a large number of spelling and punctuation errors (for example, words that after a period do not start with the first letter capitalized, subscripts that are not present, etc...) that must be corrected. The authors should do a thorough revision of the text.

2) Figure 2: I think should be moved to supplementary material.

3) Figure captions must end in a period.

4) One of the things that I miss is that the authors do not provide an evaluation of measurement errors, which should always be provided.

5) Finally, I think that some electron microscopy (SEM) images of the synthesized materials should be provided.

I believe that the paper, once these deficiencies are corrected, can be published in Materials-MDPI.

Reviewer 3 Report

The topic of the research is interesting to the scientists. Supporting data are enough but I have some observations which are listed below:

1. Methods of the research is described well but authors are requested to describe about materials in details. Exactly which form of alloys were purchased?

2. Effect of adding Sb on the optical properties of Chalcogenide Glass should be described clearly. It has been mentioned that the addition of Sb increases the refractive index (and other optical properties) but it is not mentioned that how much increased with 5 % and 10% Sb i.e. is there any correlation between percentages. Moreover, what will be happened if the amount of Sb is increased to more than 10%? Do the percentages indicate w/w?

3. There is no major mistakes observed in English but the standard of writing should be improved.  

Round 2

Reviewer 1 Report

Changes are adequate. Can be published after minor editing.

Reviewer 2 Report

The suggested changes have been made, so I consider that the manuscript can be published.